# From Bench to Bedside: Patient-Oriented Radiopharmaceutical Development in Nuclear Medicine Based on the Example of [^89^Zr]Zr-PSMA-DFO

**DOI:** 10.3390/molecules29010185

**Published:** 2023-12-28

**Authors:** Klaus Schomäcker, Felix Dietlein, Sergio Muñoz Vázquez, Feodor Braun, Thomas Fischer, Philipp Krapf, Alexander Drzezga, Markus Dietlein

**Affiliations:** 1Department of Nuclear Medicine, Faculty of Medicine and University Hospital Cologne, University of Cologne, Kerpener Str. 62, 50937 Cologne, Germany; felix.dietlein@childrens.harvard.edu (F.D.); sergio.munoz-vazquez@uk-koeln.de (S.M.V.); feodor.braun@gmail.com (F.B.); thomas.fischer@uk-koeln.de (T.F.); p.krapf@fz-juelich.de (P.K.); alexander.drzezga@uk-koeln.de (A.D.); markus.dietlein@uk-koeln.de (M.D.); 2Computational Health Informatics Program, Boston Children’s Hospital, Harvard Medical School, Boston, MA 02115, USA; 3Forschungszentrum Jülich GmbH, Institute of Neuroscience and Medicine, Nuclear Chemistry (INM-5), Wilhelm-Johnen-Straße, 52428 Jülich, Germany

**Keywords:** translational nuclear medicine, prostate carcinoma, PSMA, zirconium-89, PET

## Abstract

The interdisciplinary possibilities inherent in nuclear medicine offer an opportunity for the patient-centered development of radioactive pharmaceuticals based on specific research questions. This approach provides radiopharmaceutical manufacturers with a robust scientific foundation on which to navigate the regulatory requirements for drug approval laid down by the law. A vivid illustration of this interdisciplinary cooperation has been the development of a Zr-89-labeled PSMA ligand where reliable results have been obtained across various domains, including chemistry, radiochemistry, biochemistry, and preclinical research. This comprehensive process extended to feasibility studies conducted with carefully selected patients from a single nuclear medicine clinic. The approach demonstrates how far close collaboration between different disciplines within nuclear medicine can further the move towards patient-oriented radiopharmaceutical treatments while simultaneously meeting regulatory demands. With such a strategy, innovative radiopharmaceutical solutions can be brought to the market more swiftly and efficiently, in line with the needs of patients.

## 1. Introduction

Few medical disciplines have experienced such rapid and profound transformation in recent years as nuclear medicine. While the clinical focus of nuclear medicine physicians has long been on diagnostic imaging procedures, the emergence of various radioligands for receptors or binding sites in or on tumor cells has sparked a remarkable shift towards therapeutic approaches. Theranostics, where the synergistic combination of targeted nuclear medicine therapy and diagnostic imaging using the same molecular targets within tumor tissues has assumed an increasingly vital role, is a notable example of this shift.

The reason behind this lies in a unique aspect of nuclear medicine, its multidisciplinary nature, which necessitates collaboration among chemists, biochemists, biologists, physicists, nuclear medicine physicians, and medical professionals from various disciplines. This integration of diverse expertise guarantees a comprehensive approach to patient care, encompassing both the development of innovative radiopharmaceuticals and the precise implementation of therapeutic interventions.

As dynamic as the field of nuclear medicine may be, there are still numerous regulatory challenges to be overcome, particularly in the development of new radiopharmaceuticals. On the one hand, there seem to be conflicting guidelines regarding drug manufacturing and regulations concerning occupational safety and radiation protection. On the other hand, the extremely small batch sizes and increasingly shorter half-lives of the utilized radionuclides pose a further challenge, requiring rapid delivery and administration to patients. These factors make it exceedingly difficult to meet traditional manufacturing standards in radiopharmaceutical production.

Despite these challenges, the interdisciplinary nature of the field of nuclear medicine has made the path from bench to bedside relatively streamlined. This means that new scientific breakthroughs can bring impressive advancements in patient care more quickly. The path of a radiopharmaceutical from medical concept through to successful diagnostic and therapeutic outcomes has several stages. These are generally described in Figure 1. Some of the milestones indicated can already be reached through interdisciplinary research at any one nuclear medicine clinic. This will be discussed in detail in the following sections using the example of [^89^Zr]Zr-PSMA. However, getting the go-ahead to conduct clinical trials and eventually obtain regulatory approval is a complex and costly journey that necessitates an active partnership with the pharmaceutical industry.

## 2. Insights into the World of Radiopharmacy

The development of radiopharmaceuticals is a complex, multistage process. Some of these stages are subject to strict regulatory requirements and monitoring measures.

The process typically begins with the identification of a medical need. The role of nuclear medicine at this stage is to provide a means of early disease detection and targeted therapy.

The selection of an appropriate radionuclide that will achieve the desired imaging or therapeutic outcome is crucial here. Factors such as the physical properties, half-life, emission characteristics, and decay mode need to be carefully considered and the availability and cost of the radionuclide assessed to ensure practicality and feasibility. The feasibility of radiolabeling the chosen radionuclide is also a factor. Designing a nonradioactive precursor that will bind stably to the radionuclide is crucial. Quality control establishes the identity of radiopharmaceutical products and guarantees their purity and the safety when using them, while optimization of the radiolabeling methods maximizes reproducibility.

Biochemical and cell biological studies evaluate the affinity and specificity of candidate radiopharmaceuticals for target cells, while preclinical studies provide information on their biodistribution, pharmacokinetics, and efficacy.

Radiation physics is used to determine the best radiation dose for patients and to optimize the delivery of radiation to the target site, while a pharmacological approach is needed to transfer production to specific devices and to set up quality management and control procedures.

Clinical studies investigate the safety, efficacy, and clinical endpoints of the use of radiopharmaceuticals under consideration. Ethics approval, patient selection, safety monitoring, and data analysis are all part of this process.

Furthermore, each step of radiopharmaceutical development is subject to specific regulatory requirements to ensure a safe and effective development of radiopharmaceuticals up to the stage of marketing authorization.

The German Federal Ministry for the Environment, Nature Conservation and Nuclear Safety (BMUV) establishes the legal framework for radiopharmaceutical development. It regulates radiation protection for patients and personnel through the Federal Office for Radiation Protection (BfS). The Radiation Protection Commission (SSK) advises the BMUV and assesses the risks and benefits of various radiation applications, including radiopharmaceuticals. The German Federal Institute for Drugs and Medical Devices (BfArM) is responsible for their approval and monitoring. It evaluates quality, efficacy, and safety. Regional authorities oversee nuclear medicine facilities and enforce compliance with radiation protection regulations and operating permits. The challenging demands are summarized in Figure 1. Topics to be delved into in more detail in the following sections are indicated in this schematic diagram with headings in bold yellow letters or presented in bold black text.

## 3. What Innovative Nuclear Medicine Concept Lies behind the Development of [^89^Zr]Zr-PSMA-DFO?

In many cases of prostate cancer, the levels of prostate-specific antigen (PSA) rise again after patients have undergone surgery or radiation therapy, a phenomenon known as biochemical recurrence (BCR). Early detection of these recurring tumor lesions is crucial in order to make the right treatment decisions and improve the patient’s chances of survival.

Positron emission tomography/computed tomography (PET/CT), using radioactively labeled ligands specific to the prostate-specific membrane antigen (PSMA) is commonly employed to locate prostate cancer after BCR. This imaging procedure achieves more reliable detection rates than other imaging techniques. However, it is worth noting that PSMA PET scans fail to localize tumor lesions in approximately 20% of the patients with BCR [1]. The PSA level can also influence the likelihood of a positive PSMA-PET. A meta-analysis of 37 studies found that the detection rate of PSMA-PET in patients with a PSA level of less than 0.5 ng/mL was 36% but reached 86% in patients with a PSA level of more than 2 ng/mL. [2].

The reason for this is simple: Established PSMA tracers based on Ga-68 or F-18 are of limited use due to their short radioactive half-life, which requires PET scans to be performed within 3 h of injection. In this short time period, it is seldom possible to achieve the necessary contrasts needed for imaging, as more time is required for adequate tumor accumulation and effective clearance from the blood and surrounding tissues. Imaging based on nuclear medicine can be achieved even when the PSMA expression is comparatively low, as is the case in biochemical recurrence.

To address this, we developed a new PSMA tracer called [^89^Zr]Zr-PSMA-DFO, which has a much longer radionuclide half-life (77 h), allowing imaging to be carried out several days after injection when the background activity is minimized [3,4].

## 4. (Radio)chemistry

Radiochemical processes facilitate the binding of the radioactive Zr-89 radionuclide to the PSMA molecule, enabling doctors to track and visualize prostate cancer in the body using positron emission tomography (PET) scans.

### 4.1. Radionuclide Selection: What Is Special about Zr-89?

The radionuclide Zr-89-zirconium, with a half-life of 3.2 days, presents a valuable improvement to BCR PET imaging, particularly in combination with PET tracers labeled with short-lived isotopes like [^68^Ga]-PSMA11- or F-18-labeled PSMA ligands. Traditionally, Zr-89 has been primarily employed for labeling antibodies [5] and less so for small peptide molecules targeting PSMA (prostate-specific membrane antigen). Our innovation centers around the development of the [^89^Zr]Zr-PSMA-DFO construct for delayed PSMA-PET imaging (48 h postinjection).

This delayed imaging approach allows small tumor lesions or lesions with limited PSMA expression to be identified, as these may require more time for the PSMA inhibitor to internalize. Furthermore, it provides a means of detecting tumors that was previously overlooked due to their proximity to the ureter, following renal clearance and the associated background activity in the urine.

Zr-89 predominantly decays through positron emission (23%) and electron capture (77%) to yield the stable isotope Y-89. The emitted positrons exhibit a maximum energy of 897 keV and an average energy of 396.9 keV, which is sufficiently low to generate PET images with high spatial resolution (transverse and axial spatial resolutions at 1 cm are 4.5 and 4.7 mm for Zr-89, respectively). The potentially disruptive effects of spontaneous gamma emissions from Zr-89, emitting photons at 908.97 keV (in 99% of cases), can be counteracted by adjusting the energy window of the PET scanner [6,7,8,9].

### 4.2. The Nonradioactive Precursor: Why Specifically PSMA-DFO?

In our pursuit of a PSMA inhibitor compatible with zirconium-89, we have identified a precursor known as PSMA-DFO (EuK-2Nal-AMCH-N-sucDFO-Fe). Manufactured by ABX Radeberg in close proximity to Dresden, this compound represents an alternative to the PSMA-617 precursor initially designed for labeling with the beta-emitter Lu-177. The sole distinction between the two lies in the replacement of the DOTA chelator with Desferrioxamine (DFO). The suitability of this molecule for coupling Zr-89 to antibodies has been convincingly demonstrated [8].

However, the incorporation of zirconium-89 into the nonradioactive PSMA-617 precursor modified with DFO presented a challenge due to the presence of an iron atom at the designated site intended for the Zr-89 introduction. The iron must therefore be removed before radioactive labeling is performed. This is imperative. Conversely, commercially available [^89^Zr]Zr-oxalate can be used to label this precursor efficiently.

Reports from various research groups now describe the direct labeling of PSMA-617 with zirconium-89. This involves the chlorination of [^89^Zr]Zr-oxalate [10,11]. Some experience of its use in a clinical setting has also been documented [12,13].

## 5. Radiotracer Development/Production

### 5.1. Radiolabeling/Quality Control/Stability

Details of these features are given by our group elsewhere [3]. Figure 1 provides an overview of the chemical processes involved in the preparation and execution of the DFO-modified PSMA-617 analog.

For the radiolabeling step, the pH of a solution containing Zr-89 is adjusted using specific chemicals. Unbound Zr-89 is then removed efficiently by solid-phase extraction. To test the stability of the PSMA-targeting radiotracer, 100 μL of [^89^Zr]Zr-PSMA DFO are mixed with 1 mL of either PBS or human serum. These mixtures are kept at a constant temperature of 37 °C by means of a heating block for different time periods. Radio thin-layer chromatography is used to check the radiochemical purity of the samples at various intervals: immediately after reaching 37 °C and at 1 h, 2 h, 24 h, 48 h, and 72 h later up to 7 days. The complexes of interest stay in place on the chromatographic strip, while unbound Zr-89 moves with the mobile phase.

The labeling of [^89^Zr]Zr PSMA DFO is performed to a radiochemical purity of ≥99.9%. It shows very high stability in phosphate-buffered saline (PBS) and human serum, lasting for up to 7 days at 37 °C.

### 5.2. Optimization

The multistage production of [^89^Zr]Zr-PSMA-DFO is relatively complex (Figure 1). One particularly time-consuming aspect of the labeling process is the removal of iron from the precursor Fe-N-sucDFO-PSMA. In addition to the time factor here, there is also the question of how well this multistage process adheres to Good Manufacturing Practice (GMP) guidelines. Ideally, the actual coupling of Zr-89 to the PSMA ligand should take place using a synthesis module.

Such synthesis modules have been specifically designed to enable the versatile and efficient routine production of various radiopharmaceuticals while adhering to GMP requirements. The concept of disposable cassettes pursued in this approach allows the user to label different radionuclides on the same synthesis device without the risk of cross-contamination. The disposable cassettes include all necessary vessels, reagents, and purification columns. The automated process concludes with a final sterile filtration, including a filter integrity test. Iron removal should therefore take place before the actual labeling process. We wanted to verify whether it is possible to establish a stock of an iron-free PSMA-DFO precursor, stored in an aqueous solution at 4 °C in the refrigerator, and to determine its shelf life. This was done by investigating whether the iron-free precursor can be labeled with zirconium-89 after several months of storage without compromising the yield, cell binding, and internalization.

We were able to demonstrate that the chemical reactivity, specific binding, and internalization using the iron-free precursor remained unaffected after storage at 5 °C in an aqueous solution for a period of 6 months. This result demonstrates the feasibility of creating a stock solution of N-sucDFO-PSMA. Consequently, [^89^Zr]Zr-PSMA-DFO can be efficiently and reliably produced in accordance with GMP guidelines in a single step using an automated synthesis module.

## 6. Biochemistry/Cell Biology

The investigations primarily focused on affinity, cell binding, and internalization studies, the objective being to demonstrate that the Zr-89-PSMA ligand was at least as effective with regard to these properties as the established PSMA ligands labeled with the short-lived radionuclides F-18 and Ga-68.

### 6.1. Affinity and Binding Assay

The affinity was determined by measuring the equilibrium dissociation constant, K_D_. This is a measure of how strongly a ligand binds to its target. It quantifies the balance between the association (binding) and dissociation (release) of the ligand from its target molecule. A lower K_D_ value signifies a stronger binding affinity, indicating that the ligand interacts in a more stable manner with its target. In essence, K_D_ provides crucial information about the strength of the molecular interaction between the ligand and its target.

In our research, we assessed the binding strength between radiolabeled ligands and PSMA binding sites. To do this, we used PSMA-positive LNCaP cell lines. After the incubation of these cells under controlled conditions, varying concentrations of the radiotracer under investigation were introduced to measure their interaction with the PSMA sites.

After the incubation period, we analyzed the samples using specialized equipment and calculated the equilibrium dissociation constant (K_D_) and the maximum density of the receptors (B_max_) using advanced software. These parameters help us understand the ligand’s effectiveness in binding to PSMA, which is vital for medical imaging and diagnostics. The methodology has already been described extensively by our research group [3].

In the course of these investigations, we conducted a comparative analysis involving the Zr-89-labeled PSMA ligand and established clinical counterparts, specifically [^177^Lu]Lu-PSMA 617, [^68^Ga]Ga-PSMA 11, [^18^F]F-JK-PSMA 7, and [^99m^Tc]Tc-PSMA-I&S. The results are summarized in Table 1.

The [^89^Zr]Zr-PSMA-DFO ligand exhibited strong binding affinity to LNCaP cells, boasting a K_D_ value of 4.97 ± 0.57 nM. This affinity level was found to be on par with that of [^68^Ga]Ga-PSMA-11, [^18^F]F-JK-PSMA 7, and [^177^Lu]Lu-PSMA 617. [^99m^Tc]Tc-PSMA I&S exhibited a significantly lower affinity, approximately three times less, when compared to the other ligands.

### 6.2. Specificity of Cell Binding and Internalization

High specificity is synonymous with the targeted detection of prostate tumor lesions and enhances the diagnostic accuracy by avoiding false positives or false negatives. Additionally, it enables precise, focused therapy when using beta-emitting PSMA ligands, sparing healthy tissue. Specificity minimizes undesirable side effects and toxicities, thereby improving treatment tolerability. Lastly, it helps to avoid unnecessary examinations and interventions, ultimately increasing the efficiency and safety of prostate cancer diagnosis and therapy with radioactive PSMA ligands, thereby benefiting patients. Internalization has been a crucial parameter in assessing [^89^Zr]Zr-PSMA-DFO in relation to other PSMA ligands, as it indicates how effectively the ligand enters and interacts with the cells, which has a direct impact on its performance for diagnostic and therapeutic purposes. It also helps to determine the ligand’s ability to target and potentially treat cancer cells.

To gain insights into the specificity of our Zr-89-labeled PSMA ligand, we conducted experiments to assess its effectiveness in recognizing its specific target, PSMA (prostate-specific membrane antigen), and in entering and binding to prostate cancer cells. Our objective was to compare this performance with well-established PSMA ligands currently utilized in clinical practice.

In these experiments, we observed that cells bind to our ligand in both specific and nonspecific ways. To understand how much of this binding is specific and how much is not, we used a control substance called 2-PMPA. 2-PMPA, short for 2-(phosphonomethyl)pentanedioic acid, is a well-known inhibitor of PSMA that is frequently used in scientific studies and experiments to block PSMA activity. This is put to use when testing the specific binding of PSMA ligands and ensuring that the measured effects are indeed a result of the interaction with PSMA.

We exposed the cells to our radioactive PSMA ligands for varying periods of time (30 min, 1, 2, 3, and 5 h), both in the presence and absence of 2-PMPA. After each treatment, we removed the excess liquid, rinsed the cells with a washing solution, and then used an acid to release the radioligand bound to the cell receptors. This acid dissolved the bindings on the cell surface. Afterwards, we rinsed the cells again and measured the portion of the ligand that had entered the cells. The results for specific binding are summarized in Table 2.

To establish how much radioactivity had made its way into the cells (internalized radioactivity), we first removed the radioligands attached to the receptors by washing the cells twice with 1 mL of a 0.1 M glycine buffer solution at pH 2.8 for 5 min. We found the complexes that had originally been bound to the cell surface in this glycine wash.

We then rinsed the cells with 1 mL of PBS and measured the internalized portion by dissolving the cells in 1 mL of 1 M NaOH. The internalized portion was recovered in the NaOH fraction.

The degree of cell binding was calculated by measuring the radioactivity in both the glycine and NaOH fractions. The internalized portion, expressed as a percentage of the cell binding, showed how much radioactivity had entered the cells in relation to the amount bound to them. This indicated how well our ligand had worked and how it compared to other well-established ligands. The results regarding internalization are summarized in Table 3.

### 6.3. Interim Summary

In all the in vitro parameters examined, including affinity (K_D_ values), specific cellular binding, and internalization, the results for the newly developed Zr-89 ligand closely mirrored those obtained with established radioactive PSMA ligands. The exception here was [^99m^Tc]Tc-PSMA I&S, especially in terms of the K_D_ value.

At this point on our journey “from bench to bedside”, we found ourselves at a critical juncture, as we needed to decide whether to continue the comparison of Zr-89 with the established reference tracers in a living organism, specifically in animal experiments. To minimize the number of experimental animals used, we decided to carry out further comparisons with the F-18- and Ga-68-labeled PSMA ligands alone.

## 7. Preclinical Research

### 7.1. Animal Experiments: Tumor/Background Ratios

An overarching aspect of animal experiments is the evaluation of the tolerance to and potential side effects of the candidate radiopharmaceutical under investigation.

In the animal-based segment of the study, our specific aim was to test the hypothesis that [^89^Zr]Zr-PSMA DFO could achieve higher tumor-to-background ratios than the short-lived tracers under investigation.

The term “tumor-to-background ratio” refers to the ratio of radioactive signal intensity or radioactivity concentration within the tumor (the target region) compared to the surrounding area, often referred to as the “background”.

A higher tumor-to-background ratio means that the radioactive signal within the tumor is more pronounced and distinct than in the surrounding normal or background tissues. This is crucial for the imaging quality, as it enhances the ability to detect and locate the tumor accurately. Where the tumor-to-background ratio is low, identifying the tumor can become a challenge, rendering imaging less reliable.

The aim of our investigation was to determine whether the Zr-89-labeled PSMA ligand can generate a higher tumor-to-background ratio than other tracers. The higher this ratio is, the more precise and reliable the imaging will be, and this is of paramount importance for the diagnosis and monitoring of cancer.

Details of how these investigations were conducted and the results obtained have been published extensively elsewhere [3]. We will therefore provide just a brief summary of the experimental procedures used and highlight the key findings. Among the animal-based studies, both biodistribution in animal tissue and small animal PET scans in mice bearing a LNCaP tumor xenograft were conducted.

Biodistribution studies assessed radioactivity accumulation in various tissues and calculated the tumor-to-blood, tumor-to-muscle, and tumor-to-kidney ratios. In adherence to EU directive 2010/63/EU and regional authorities’ approval, only 35 male CB17-SCID mice (age: 6 weeks, weight: 17–20 g) were used. Tumor cells were implanted subcutaneously into the right flank, and once the tumors had reached 300 to 600 mg, the mice were sacrificed for biodistribution studies or PET imaging.

On the experiment day, each mouse received intravenous injections of [^89^Zr]Zr-PSMA-DFO, [^18^F]F-JK-PSMA-7, or [^68^Ga]Ga-PSMA-11. The mice were sacrificed at specified time points, and their organs (blood, liver, spleen, kidneys, muscle, bone, thyroid, lungs, intestines, tumor, heart, and prostate) were dissected and weighed and radioactivity measured. The biodistribution studies were conducted at 2, 4, and 24 h after the injection of [^89^Zr]Zr-PSMA-DFO, [^18^F]F-JK-PSMA-7, or [^68^Ga]Ga-PSMA-11. These time intervals were chosen to accommodate the short physical half-lives of Ga-68 and F-18 while simultaneously comparing the performance of the Zr-89-labeled PSMA ligand with those labeled with short-lived radionuclides. Furthermore, these intervals highlight the superiority of the Zr-89-labeled tracer after an extended administration period (24 h). The results are expressed as a percentage of the injected dose per gram of tissue (% ID/g) (*n* = 5). After 24 h, [^89^Zr]Zr-PSMA-DFO displayed much higher ratios between the tumor and the surrounding areas (tumor/blood: 309 ± 89, tumor/muscle: 450 ± 38) in LNCaP tumor xenografts when compared to [^68^Ga]Ga-PSMA-11 (tumor/blood: 112 ± 57, tumor/muscle: 58 ± 36) or [^18^F]F-JK-PSMA-7 (tumor/blood: 175 ± 30, tumor/muscle: 114 ± 14) after only 4 h (*p* < 0.01).

### 7.2. Animal Experiments: Small Animal PET

After the biodistribution experiments, we used PET scans to confirm the biodistribution findings. For this purpose, PET scans were obtained of LNCaP tumors in mice with [^68^Ga]Ga-PSMA-11 and [^18^F]F-JK-PSMA-7 tracers, each mouse receiving 10 MBq of the tracer.

We also checked to see whether [^89^Zr]Zr-PSMA-DFO specifically binds to PSMA by using a PSMA blocker called 2-PMPA. Some mice received 2-PMPA, while some did not. All scans were done while the mice were under anesthesia.

The scans lasted for 60 min and began 60 min after the tracer injection. Further scans were carried out at 4 h, 21 h, and 48 h after the injection of [^89^Zr]Zr-PSMA-DFO. These scans helped us to see where the tracer went in the body. The summed images were reconstructed using an iterative OSEM3D/MAP procedure resulting in voxel sizes of 0.47 × 0.47 × 0.80 mm. Postprocessing and image analyses were performed with VINCI 4.72 (Max-Planck-Institute for Metabolism Research, Cologne, Germany). The images were Gauss-filtered (1 mm FWHM) and intensity-normalized to the injected dose, corrected for body weight (SUV_bw_). Finally, we adjusted the images based on the amount of tracer used in relation to the weight of the mouse and thereby achieved better results.

In live, small animal PET imaging, we observed that the ability to visualize tumors with [^89^Zr]Zr-PSMA-DFO was similar to that achieved with [^68^Ga]Ga-PSMA-11 or [^18^F]F-JK-PSMA-7 at early time points (1 h after injection). Furthermore, PET scans performed up to 48 h after injection continued to provide clear tumor visualization, even at later time points [3].

### 7.3. Toxicity

To rule out acute toxicity of the nonradioactive precursor, we commissioned a certified external laboratory to test this. A group of 10 mice were injected intravenously with 1 mg/kg body weight of the nonradioactive precursor EuK-2Nal-Amc-N-sucOf-Fe (PSMA-DFO). None of the animals exhibited any pathological signs or clinical symptoms attributable to administration of the PSMA precursor over a 14-day observation period. There were no fatalities from a single dose of 1 mg/kg body weight, and the examination of the organs for visible changes yielded no significant findings. As a result, it was concluded that the lethal dose (LD**_50_**) in mice exceeds 1 mg/kg body weight.

### 7.4. Interim Summary

Preclinical data from mice with LNCaP tumor xenografts have been quite promising and indicate that, compared to the widely used PSMA tracers [^68^Ga]Ga-PSMA-11 and [^18^F]F-JK-PSMA-7, [^89^Zr]Zr-PSMA-DFO exhibits a higher tumor-to-background ratio thanks to its remaining in the tumor tissue for a longer period after injection. Furthermore, after 48 h, there was only minimal radioactive background radiation in the kidneys and bladder.

Based on these promising preclinical findings, we decided to initiate a feasibility study in patients with biochemical recurrence. All of our working hypotheses were confirmed.

## 8. First Feasibility Study: PET/CT on Patients

In light of the encouraging results from our preclinical investigations, we decided to pioneer the use of [^89^Zr]Zr-PSMA-DFO for PET imaging in prostate cancer patients experiencing biochemical recurrence. Our aim was to pinpoint tumor lesions suitable for metastasis-directed therapy (MTD) or salvage radiotherapy (S-RT) in 14 patients who previously had negative PET scan results using conventional PSMA tracers.

As the study has been published at length elsewhere [4], we will provide just a brief overview of the methodology and key findings.

The collection of patient data involves a retrospective analysis of clinical data from a feasibility study and does not constitute a clinical trial under the German Medicines Act. The Institutional Review Board has granted approval for this study and the utilization of data for a retrospective analysis. All participants have provided their consent by signing a written informed consent form, allowing PET imaging and permitting the use of their data for a retrospective analysis. All procedures were conducted in compliance with the regulations of the responsible local authorities (District Administration of Cologne, Germany). Specifically, this falls within the scope of § 13 paragraph 2b of the German Medicines Act. This provision allows physicians to manufacture drugs without a permit if the following conditions are met:
The drugs are manufactured under their immediate professional responsibility, andthese drugs are exclusively intended for personal use in a specific, named patient, ensuring that the manufacturing and administering physician has examined the patient beforehand and, consequently, has determined the indication for the respective drug.


If radioactive drugs were to be investigated as part of a clinical trial, an explicit pharmaceutical manufacturing permit would be required under drug regulations.

In 8 out of 14 patients who initially had negative PET scans using the usual tracers, [^89^Zr]Zr-PSMA-DFO was able to detect 15 PSMA-positive lesions. Among these eight patients, the new scans showed that some had a recurrence of their prostate cancer in the same area as before (three out of eight), while others had cancer that had spread to nearby lymph nodes (three out of eight) or even to distant sites in the body (two out of eight).

Based on these findings, five of the patients with detected lesions received targeted radiation therapy, two received hormone therapy, and one did not receive any treatment. The validity of 14 out of 15 detected lesions was supported either by examining the tissue under a microscope (histology), monitoring the patients’ health after radiation therapy, or performing additional imaging tests. When we compared the 15 lesions detected by [^89^Zr]Zr-PSMA-DFO with images on the original PET scans, we found that the standard tracers did show some signals in 7 out of 15 lesions, but it was too faint to confidently identify them. The level of radioactivity in the 15 lesions detected by [^89^Zr]Zr-PSMA-DFO (measured as SUVmax) was significantly higher than that seen on the original PET scans.

In the meantime, another research group has also published their findings with a very similar Zr-89-labeled PSMA ligand in patients with biochemical recurrence [14]. This research team shares our view that Zr-89-labeled PSMA ligands appear to be effective in detecting prostate cancer in men with BCR and low PSA levels that may not be visible on scans using PSMA ligands labeled with short-lived radionuclides like Ga-68. The higher detection rates and lesion-to-background ratios observed in the 48 h scans, as compared to the 24 h ones, suggest that imaging at a later time point may be preferable.

## 9. Radiation Physics/Dose

Ionizing radiation can cause direct DNA damage in human cells, in particular, single-strand and double-strand breaks. It can also cause indirect damage by generating reactive oxygen species (ROS). Both of these processes result in mutations and alterations in oncogenes and tumor suppressor genes, thereby promoting the development of late-onset malignancies even at dose levels commonly used in diagnostic procedures. This is why dose calculations are of critical importance whenever radioactive substances are being applied.

From a radiation protection perspective, Zr-89 is not the ideal radionuclide for labeling PSMA ligands. Its half-life of 3.27 days inevitably results in a higher radiation exposure for the patient compared to that caused by shorter-lived positron emitters like Ga-68 (68 min) or F-18 (110 min). Zr-89 also poses challenges with regard to the management of both liquid and solid radioactive waste. However, in weighing up the balance between benefit and risk, the scales tip in favor of benefits. The theoretical, deferred-to-later-in-life risk can be considered secondary in view of the gain in both lifespan and quality of life achieved through timely, customized, patient-centric therapy based on tumor lesions diagnosed with Zr-89-labeled PSMA ligands, which might have been overlooked with shorter-lived tracers.

Most of the data on radiation exposure come from studies using Zr-89-labeled antibodies in patients. The critical organs in these studies are the liver and spleen. The results published for these organs range from 1 mSv/MBq to a maximum of 2.6 mSv/MBq. The effective dose estimates for Zr-89-labeled antibodies lie between 0.2 and 0.6 mSv/MBq [15,16,17,18,19,20,21,22,23,24,25].

Recent dose information for Zr-89-labeled PSMA ligands has now been published. Our own paper gives a dose estimate of 3.3 mSv/MBq for the kidneys, the organ considered most critical in this case [4]. Rosar et al. reported that the parotid gland received the highest absorbed dose of [^89^Zr]Zr-PSMA-617 at 0.601 ± 0.185 mGy/MBq, followed by the kidneys at 0.517 ± 0.125 mGy/MBq and the submandibular gland at 0.468 ± 0.136 mGy/MBq [14].

There is currently limited data available regarding the degree to which personnel in nuclear medicine clinics are exposed to radiation during examinations using Zr-89 tracers. Alfuraih et al. compared the measurements taken to those that would occur when using the routinely employed [^18^F]F-deoxyglucose for PET scans. At a distance of 1 m from the patient, the authors measured a dose rate of 0.143 μSv MBq-1 h-1 for F-18 and of 0.154 μSv MBq-1 h-1 for Zr-89 [26].

## 10. Where We Now Stand and What Is Left to Accomplish

### 10.1. Summary of the Current Radiochemical Preclinical and Initial Clinical Findings in Radiopharmaceutical Development

[^89^Zr]Zr-PSMA-DFO has been developed as a radioligand combining several favorable characteristics. With satisfactory tumor affinity, there is a sufficient clearance of radioactivity from the blood and tissues within 48 h, which could otherwise overshadow the radioactivity accumulated in the tumor. After 48 h, hardly any detectable radioactivity remains, even in the bladder. Zirconium is a residualizing radionuclide, because it remains trapped inside the target cells even after degradation of the radiolabeled biomolecules that transport it [3]. Thus, Zr-89 remains adequately retained in the tumor even after 48 h, meaning that significant tumor-to-background ratios can be obtained on administration of the radioligand at 48 h intervals. In this way, tumors can be detected on PET/CT scans, something that cannot be achieved with PSMA ligands labeled with short-lived radioisotopes.

### 10.2. Post-Diagnosis Observation of Patients following the Unexpected Zr-89 Diagnosis and Therapeutic Implications

The PET/CT with [^89^Zr]Zr-PSMA-DFO was offered to 14 patients to identify patients with the option of local treatment, e.g., salvage radiotherapy of the prostate fossa or metastasis-directed therapy in the case of oligometastasis. Eight of the fourteen patients revealed one or more [^89^Zr]Zr-PSMA-positive lesions. Five of the eight PSMA-positive patients received salvage radiotherapy (three patients: prostate fossa) or metastasis-directed therapy (one patient: PSMA-positive lymph nodes, one patient: bone marrow metastasis and lymph node metastasis). Another two patients with [^89^Zr]Zr-PSMA-DFO-positive PET scans received androgen deprivation therapy, since they were not eligible for MDT (too many [^89^Zr]Zr-PSMA-positive metastases or too short a doubling time of the PSA level). One patient with a PSMA-positive coin lesion in the lung exhibited stable disease in a follow-up CT after 7 months, and their PSA levels remained stable for 11 months, so that the urologists pursued watchful waiting for this patient.

The five patients who underwent salvage radiotherapy or metastasis-directed therapy revealed a decrease in their PSA level as their biochemical response to the therapy. During follow-up care, two patients revealed a continuous long-term remission of PSA levels without androgen deprivation on their last visits at 38 months and 45 months, respectively, after salvage radiotherapy. Two patients with an initial decrease in PSA level after salvage radiotherapy or the radiotherapy of lymph node metastases showed the same PSA level as was shown at the time of the [^89^Zr]Zr-PSMA-DFO PET/CT after 32 months and 42 months, respectively. The biochemical relapse was explained by PSMA-positive lymph node metastases in these two patients. The patient who underwent radiotherapy of the [^89^Zr]Zr-PSMA-positive bone metastasis had lymph node dissection 9 months later due to an increase in the PSA level and one [^89^Zr]Zr-PSMA-positive iliac lymph node. Lymph node metastases were confirmed histologically in 2 out of 22 lymph nodes. Over a follow-up period of 42 months, the PSA level remained lower than at the time of the [^89^Zr]Zr-PSMA-DFO PET/CT without androgen deprivation.

### 10.3. Potential of the Investigated Radiopharmaceutical

The high long-term stability of [^89^Zr]Zr-PSMA-DFO with a half-life of 3.3 days has practical advantages: There is no need to produce the radiopharmaceutical several times a week, and the time pressure of manufacturing Ga-68- or F-18-labeled PSMA ligands, along with the radioactivity loss during transport, is eliminated. While F-18 tracers are readily available through existing networks, the fact that [^89^Zr]Zr-PSMA-DFO can be stored longer and used with minimal effort seven days after synthesis speaks for itself. Furthermore, it can be transported over long distances without significant quality loss.

However, the [^89^Zr]Zr-PSMA-DFO tracer cannot compete with short-lived PSMA tracers but serves as a valuable addition for patients with low PSMA expression or rising PSA levels and negative PET scans with short-lived PSMA tracers. Clinical trials should therefore be considered for a well-balanced assessment in this specific patient population.

### 10.4. Next Steps in the Clinical Establishment of the Radiopharmaceutical

The dedicated constellation of inclusion criteria (biochemical relapse after prostatectomy and salvage radiotherapy, PSMA negativity with Ga-68 or F-18-labeled PSMA ligands, and PSA level of more than 0.5 ng/mL) will require multicentric recruitment into a clinical trial. In our retrospective first-in-human study, the readers could revise their initial interpretation of the [^68^Ga]Ga-PSMA-11 PET or the F-18-labeled PSMA PET scans in ambiguous cases on the basis of the [^89^Zr]Zr-PSMA-DFO PET scan. This clinically driven strategy might have increased the sensitivity of the [^89^Zr]Zr-PSMA-DFO PET. In a prospective clinical trial, the [^89^Zr]Zr-PSMA-DFO PET scans should be evaluated in an unbiased manner, progressing solely from the information that a prior PET scan with [^68^Ga]Ga-PSMA-11 or [^18^F]F-JK-PSMA-7 was read as negative. As we see it, a randomized use of [^68^Ga]Ga-PSMA-11, F-18-labeled PSMA, or [^89^Zr]Zr-PSMA-DFO would not meet the clinical challenge.

To conduct a clinical trial with [^89^Zr]Zr-PSMA-DFO in accordance with the German Medicines Act, as mentioned above, it is crucial to obtain a manufacturing license for the radiopharmaceutical. This involves manufacturing control and quality assurance through a quality management system, including all the steps outlined briefly in Figure 1 under “Pharmacy”. These measures must meet all the requirement of the government regulatory agencies.

## Data Availability

The data presented in this study are openly available in Translational Development of a Zr-89-Labeled Inhibitor of Prostate-specific Membrane Antigen for PET Imaging in Prostate Cancer|Molecular Imaging and Biology (springer.com).

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
