# Peer review of "From Bench to Bedside: Patient-Oriented Radiopharmaceutical Development in Nuclear Medicine Based on the Example of [^89^Zr]Zr-PSMA-DFO"

_molecules, 2023, doi:10.3390/molecules29010185_

Round 1

Reviewer 1 Report

Comments and Suggestions for Authors

Schomacker et al provide a review article regarding the development of 89Zr-PSMA-DFO.  Overall, this is a comprehensive review of the rationale for development, preclinical and clinical studies for the compound.

My main comment for improvement is overall organization.  I appreciate their presentation of 8 separate development concepts needed for this.  Mostly, the paper subdivides along these sections.  While I appreciate that this organization matches the proposed Table/concept, it is not easy to follow.  I would suggest separately organizing the paper in more easily understandable sections but can certainly still present the development concepts.

Comments on the Quality of English Language

Overall ok, some improvements on flow of language particular in introduction and abstract can be made.

Author Response

Referee 1

My main comment for improvement is overall organization.  I appreciate their presentation of 8 separate development concepts needed for this.  Mostly, the paper subdivides along these sections.  While I appreciate that this organization matches the proposed Table/concept, it is not easy to follow.  I would suggest separately organizing the paper in more easily understandable sections but can certainly still present the development concepts.

In response to your valuable feedback, for which we sincerely thank you, we extensively discussed your objection within the working group. As a result, we have decided to structure section 10 for better clarity into four subsections with the following headings:

10.1 Summary of the current radiochemical preclinical and initial clinical findings in radio-pharmaceutical development.

10.2 Post-diagnosis observation of patients following the unexpected Zr-89 diagnosis and therapeutic implications

10.3 Potential of the investigated radiopharmaceutical

10.4 Next steps in the clinical establishment of the radiopharmaceutical

Reviewer 2 Report

Comments and Suggestions for Authors

The manuscript provides a case study for the translation of a radiopharmaceutical from discovery to Phase 0/1 clinical trials in humans.

The manuscript is marked as a review, but most of the experimental data discussed comes from the authors' previous two publications, cited as ref. 3 and 4, with the other references providing some background information about the chemistry of 89Zr, its use of for PET imaging etc. Evaluation of a structurally similar 89Zr-labeled PSMA derivative in humans is also briefly mentioned (ref 12-14) and the results are compared to the authors' own.

The aim of the manuscript is apparently to use the concrete case of 89Zr-PSMA-DFO development to list the key milestones and challenges in the clinical translation of a novel diagnostic radiopharmaceutical in the regulatory environment of Germany.

I believe that the manuscript can be published after the following issues are addressed:

1. Which clinical trial phase does the clinical data presented in the manuscript for 89Zr-PSMA-DFO correspond to?
The last section discusses plans for future clinical trials. Which clinical phase are you aiming for? The last paragraph discusses the need to obtain a manufacturing license to conduct clinical trials - exact requirements are phase-dependent, so it's important to specify the phase

2. Figures and tables illustrating the evaluation of 89Zr-PSMA-DFO end at cell internalization studies. Why not provide reproductions of animal or human scans to illustrate the sensitivity of the tracer in metastasis detection and and its uptake in dose-limiting organs (e.g. kidneys)?

3. Page 8, lines 317-318: "Details of how these investigations were conducted and the results obtained have been published extensively elsewhere."

Please provide a reference to a publication where the discussed data can be found. I assume it should be ref. 3?

4. Page 9, lines 364-365: "it 364 was concluded that the lethal dose (LD50) in mice exceeds 1 mg/kg body weight"

What about the No Observed Adverse Effect Level dose? Is this not much more relevant for clinical translation than LD50?

5. Page 3, lines 125-127. "we developed a new PSMA tracer called [89Zr]Zr-PSMA-DFO, which has a much longer half-life (77 hours)"

Is this is the biological half-life of the ligand or the decay half-life of 89Zr? If it's the latter, then the wording needs to be corrected.

6. Page 4, lines 146-147. "The emitted positrons exhibit a maximum energy [...], which is sufficiently low to generate PET 147 images with high spatial resolution."

Please provide an estimate of typical spatial resolution for 89Zr imaging rather than just a range of positron energies.

7. Page 8, lines 322-327

Please mention the timepoints at which the biodistribution was assessed, and explain their choice.

Is the number of the mentioned EU directive (2010/60/EU) correct? Googling for the content shows that the directive with this number is about "certain derogations for marketing of fodder plant seed mixtures intended for use in the preservation of the natural environment"

Why is the tumor size reported in mg and not in cubic mm? It seems that, in the cited work, it was the volume of the tumor that was measured, rather than the mass.

8. Page 8, lines 348-349 "We used a 348 special program to correct the images and improve their clarity."

Please provide the name of the (software?) program and its producer.

9. Page 9, lines 401-402 "This research team shares our view that [89Zr]Zr-PSMA-617 PET/CT appears to be effective in detecting prostate cancer in men with BCR and low PSA levels"

The meaning of this sentence is unclear to me. The current manuscript focuses on the advantages of 89Zr-PSMA-DFO over 68Ga and 18F labeled analogs. This sentence suddenly switches to 89Zr-PSMA-617, while there has been minimal discussion of the actual results obtained for this tracer.

10. Page 3, line 129 "The radiochemical facilitates attachment of the radioactive Zr-89 label to the PSMA 129 molecule"

This sentence apparently contains a typo

11. Page 4, lines 133-134 "The radionuclide Zr-89-zirconium [...] presents a valuable improvement to BCR PET imaging, particularly in combination with short-lived PET tracers"

Did you mean "in comparison with short-lived PET tracers"?

Also, please keep the distinction between biological half-life and isotope decay half-life. The wording here should probably be changed to "PET tracers labeled with short-lived isotopes" 

12. Scheme 1: Harmonize capitalization and remove a hanging (empty) bulletpoint in the "Radiation physics/Dose" section

Comments on the Quality of English Language

The use of some words seems strange to me. For example: "Zr-89 remains adequately stored in the tumor". Why not "retained"?

"This was done by asking whether the iron-free precursor 200 can be labeled with zirconium-89" Why not "investigating"?

Author Response

Referee 2

  1. Which clinical trial phase does the clinical data presented in the manuscript for 89Zr-PSMA-DFO correspond to?

The last section discusses plans for future clinical trials. Which clinical phase are you aiming for? The last paragraph discusses the need to obtain a manufacturing license to conduct clinical trials - exact requirements are phase-dependent, so it's important to specify the phase

We appreciate this valuable input and would like to respond as follows. All subsequent changes will be written in the manuscript in bold and italics.

The collection of patient data involves a retrospective analysis of clinical data from a feasibility study and does not constitute a clinical trial under the German Medicines Act. The Institutional Review Board has granted approval for this study and the utilization of data for a retrospective analysis. All participants have provided their consent by signing a written informed consent form, allowing PET imaging, and permitting the use of their data for a retrospective analysis. All procedures were conducted in compliance with the regulations of the responsible local authorities (District Administration of Cologne, Germany). Specifically, this falls within the scope of § 13 paragraph 2b of the German Medicines Act. This provision allows physicians to manufacture drugs without a permit if the following conditions are met:

  • The drugs are manufactured under their immediate professional responsibility, and
  • these drugs are exclusively intended for personal use in a specific, named patient, ensuring that the manufacturing and administering physician has examined the patient beforehand and, consequently, has determined the indication for the respective drug.

If radioactive drugs were to be investigated as part of a clinical trial, an explicit pharmaceutical manufacturing permit would be required under drug regulations.

  1. Figures and tables illustrating the evaluation of 89Zr-PSMA-DFO end at cell internalization studies. Why not provide reproductions of animal or human scans to illustrate the sensitivity of the tracer in metastasis detection and and its uptake in dose-limiting organs (e.g. kidneys)?

Thank you for the suggestion. At this juncture, we would like to note that all pertinent figures are already available in our cited previous works.

  1. Page 8, lines 317-318: "Details of how these investigations were conducted and the results obtained have been published extensively elsewhere."

Please provide a reference to a publication where the discussed data can be found. I assume it should be ref. 3?

Thank you for the suggestion. We have included Reference 3.

  1. Page 9, lines 364-365: "it was concluded that the lethal dose (LD50) in mice exceeds 1 mg/kg body weight"

What about the No Observed Adverse Effect Level dose? Is this not much more relevant for clinical translation than LD50?

Firstly, thank you for providing a more detailed perspective on the results. To the best of our knowledge, the no-observed-adverse-effect-level (NOAEL) is an integral component of non-clinical risk assessment (Reference: Michael A. Dorato, Jeffery A. Engelhardt, "The no-observed-adverse-effect-level in drug safety evaluations: Use, issues, and definition(s)," Regulatory Toxicology and Pharmacology, Volume 42, Issue 3, 2005, Pages 265-274, https://doi.org/10.1016/j.yrtph.2005.05.004). It represents a professional opinion grounded in the study's design, drug indication, expected pharmacology, and the spectrum of off-target effects. It's important to note that there is no consistent standard definition for NOAEL.

In the toxicity study commissioned by us, the focus was on acute toxicity. Since no adverse effects or tissue changes were observed in a sufficient number of animals at 1 mg/kg body weight, we (as a precautionary measure) agreed on stating an LD50 > 1 mg/kg body weight. More precise information regarding side effects is ascertainable only within the context of a qualified clinical trial. Therefore, we do not intend to delve deeper into this point in the text.

  1. Page 3, lines 125-127. "we developed a new PSMA tracer called [89Zr]Zr-PSMA-DFO, which has a much longer half-life (77 hours)"

Is this is the biological half-life of the ligand or the decay half-life of 89Zr? If it's the latter, then the wording needs to be corrected.

This refers to the physical half-life of Zr-89. We have incorporated this crucial addition into the text.

  1. Page 4, lines 146-147. "The emitted positrons exhibit a maximum energy [...], which is sufficiently low to generate images with high spatial resolution."

Please provide an estimate of typical spatial resolution for 89Zr imaging rather than just a range of positron energies.

We have made the following addition in the text within parentheses: transverse and axial spatial resolutions at 1 cm are 4.5 and 4.7 mm for Zr-89, respectively.

  1. Page 8, lines 322-327

Please mention the timepoints at which the biodistribution was assessed, and explain their choice.

Thank you for this suggestion. The biodistribution studies were conducted at 2, 4, and 24 hours after the injection of [89Zr]Zr-PSMA-DFO, [18F]F-JK-PSMA-7, or [68Ga]Ga-PSMA-11. These time intervals were chosen to accommodate the short half-lives of Ga-68 and F-18 while simultaneously comparing the performance of the Zr-89-labeled PSMA ligand with those labeled with short-lived radionuclides. Furthermore, these intervals highlight the superiority of the Zr-89-labeled tracer after an extended administration period (24 hours).

We have inserted the passage into the manuscript.

Is the number of the mentioned EU directive (2010/60/EU) correct? Googling for the content shows that the directive with this number is about "certain derogations for marketing of fodder plant seed mixtures intended for use in the preservation of the natural environment"

Thank you for this valuable correction. It should indeed be 2010/63/EU. We have rectified this error in the manuscript.

Why is the tumor size reported in mg and not in cubic mm? It seems that, in the cited work, it was the volume of the tumor that was measured, rather than the mass.

In comparable publications, it is customary to express the results as a percentage of the injected dose per gram of tissue (% ID/g). Therefore, specifying the tumor volume is not conducive to our objective.

  1. Page 8, lines 348-349 "We used a 348 special program to correct the images and improve their clarity."

Please provide the name of the (software?) program and its producer.

We had omitted these details to avoid overwhelming the readers. Following your specific inquiry, we have now added the following passage to the manuscript:

"Summed images were reconstructed using an iterative OSEM3D/MAP procedure resulting in voxel sizes of 0.47×0.47×0.80 mm. Post-processing and image analysis were performed with VINCI 4.72 (Max-Planck-Institute for Metabolism Research, Cologne, Germany). Images were Gauss-Filtered (1 mm FWHM) and intensity-normalized to injected dose, corrected for body weight (SUVbw)."

Page 9, lines 401-402 "This research team shares our view that [89Zr]Zr-PSMA-617 PET/CT appears to be effective in detecting prostate cancer in men with BCR and low PSA levels"

The meaning of this sentence is unclear to me. The current manuscript focuses on the advantages of 89Zr-PSMA-DFO over 68Ga and 18F labeled analogs. This sentence suddenly switches to 89Zr-PSMA-617, while there has been minimal discussion of the actual results obtained for this tracer.

Thank you for pointing out the unclear wording. We intended to convey that the results achievable with [89Zr]Zr-PSMA-617 PET/CT are comparable to those attainable with [89Zr]Zr-PSMA-DFO. This is not surprising given the chemical similarity of the substances. We have revised the text as follows: “This research team shares our view that Zr-89 labeled PSMA-Ligands appear to be effective in detecting prostate cancer in men with BCR and low PSA levels that may not be visible on scans using PSMA ligands labeled with short-lived radionuclides like Ga-68.”

  1. Page 3, line 129 "The radiochemical facilitates attachment of the radioactive Zr-89 label to the PSMA 129 molecule"

This sentence apparently contains a typo

Thank you for the note. We have modified the sentence as follows: “Radiochemical processes facilitate the binding of the radioactive Zr-89 radionuclide to the PSMA molecule, enabling doctors to track and visualize prostate cancer in the body using positron emission tomography (PET) scans.”

  1. Page 4, lines 133-134

"The radionuclide Zr-89-zirconium [...] presents a valuable improvement to BCR PET imaging, particularly in combination with short-lived PET tracers" Did you mean "in comparison with short-lived PET tracers"? Also, please keep the distinction between biological half-life and isotope decay half-life. The wording here should probably be changed to "PET tracers labeled with short-lived isotopes"

Yes, you are correct. We have amended the corresponding passage in the text.

  1. Scheme 1: Harmonize capitalization and remove a hanging (empty) bulletpoint in the "Radiation physics/Dose" section Comments on the Quality of English Language The use of some words seems strange to me. For example: "Zr-89 remains adequately stored in the tumor". Why not "retained"? "This was done by asking whether the iron-free precursor 200 can be labeled with zirconium-89" Why not "investigating"?

Thank you for your feedback; we have made the necessary corrections.